# Psychotic Symptoms in Cataract Patients Without Overt Psychosis Are Ameliorated Following Successful Cataract Surgery

**DOI:** 10.3390/diseases13070224

**Published:** 2025-07-18

**Authors:** Georgios D. Floros, Ioanna Mylona, Stylianos Kandarakis

**Affiliations:** 12nd Department of Psychiatry, Aristotle University of Thessaloniki, 541 24 Thessaloniki, Greece; 2Department of Ophthalmology, General Hospital of Serres, 621 00 Serres, Greece; milona_ioanna@windowslive.com; 3First Department of Ophthalmology, General Hospital “G. Gennimatas”, National and Kapodistrian University of Athens, 115 27 Athens, Greece; s.kandarakis@gmail.com

**Keywords:** psychosis, paranoid ideation, psychotic symptoms, cataract, phacoemulsification

## Abstract

Background: Cataract is the leading cause of severe, non-traumatic vision loss worldwide, leading to multiple adverse outcomes in mental health, including depression, anxiety, and cognitive decline; however, the relationship to psychotic symptoms remains unclear. While congenital vision loss appears protective against psychosis, acquired vision loss or acute deprivation are inducing psychotic symptoms. Methods: This study of 200 consecutive cataract patients, with severe vision loss, compares Paranoid Ideation and Psychoticism symptoms pre surgery, measured with the SCL-90-R scale, to those symptoms that persisted two months post-surgery. Results: The results confirm the hypothesis that cataract surgery is associated with a reduction in those symptoms (Wilcoxon Z = 5.425, *p* < 0.001 for Paranoid Ideation and Wilcoxon Z = 6.478, *p* < 0.001 for Psychoticism). Higher improvement in those variables was associated with higher improvement in visual acuity while controlling for age, gender and stressful life events during the past six months. Conclusions: Those results point to the importance of addressing loss of visual function especially in patients with pre-existing psychotic symptoms or signs of cognitive decline.

## 1. Introduction

Cataract was the leading global cause of blindness in those aged 50 years and older in 2020 [1]. Significant vision loss resulting from untreated cataracts has a considerable negative effect on both health quality and mental well-being, with cataract patients repeatedly demonstrating increased rates of depression and anxiety compared to the general population [2,3]. A meta-analysis of published materials found that, in six controlled studies, the reduction in depression was higher in the surgery group than in the control group [4]. Additionally, this meta-analysis found that cognitive function improved significantly after surgery.

While the effect of cataract surgery on those with mental health issues appears validated by the research results, much less is clear with regard to psychotic symptomatology. Although there is a continuous debate on the relationship between overt psychosis and the loss of visual function, this debate mostly revolves on whether major psychotic illnesses may be associated on a neurobiological level with congenital blindness [5]. Recently, Pollak and Corlett noted that visual experience is critical in the construction of our internal world model and the functionality of this model may become severely disrupted following visual loss. [6]. The authors have offered a computational model that can explain how congenital visual loss protects against psychosis, while later-life visual loss predisposes towards it, with several hypotheses that remain to be tested in the congenitally blind. Congenitally blind subjects presumably are protected by psychosis due to visual loss-induced changes in NMDA receptor structure happening over an appropriately long period of developmental time, while visual loss in later life cannot put the same mechanisms in motion, since brain function has been pre-routed. The longitudinal study that would validate whether this association does exist requires vast resources [7].

However, a much clearer picture emerges in later life, where the loss of visual function in cerebral visual syndromes is associated with hallucinations, either with reduced insight, as in Anton syndrome, with insight, as in the Charles Bonnet Syndrome, or associated with delusions, as in Capgras and Fregoli syndromes [8]. Sensory and, especially, visual deprivations have also long been associated with psychotic symptoms in previously unaffected subjects [9,10]. Those symptoms develop over a short period of sensory deprivation, with complete visual deprivation leading to hallucinations in the span of five consecutive days [10]. A recently published, cross-sectional analysis of data from the 2014 UK Adult Psychiatric Morbidity Survey [11] found a high association for visual impairment and a positive categorization on the Psychosis Screening Questionnaire (PSQ). Although the positive categorization in the PSQ relates to possible diagnosis of a psychotic disorder, out of the 7107 contributors, only 8 (<0.1%) reported having ever been diagnosed with psychosis or schizophrenia. Unfortunately, the categorization of visual impairment and psychotic symptoms was dichotomous, and few meaningful conclusions could be extracted. The data regarding less pronounced psychotic symptomatology, defined as psychotic-like symptoms and experiences, are scarce.

The definition and assessment of psychotic-like symptoms remain challenging [12]. A key aspect in the categorization of psychotic-like symptoms is the distinction between transient perceptual disturbances, prodromal symptoms of psychosis, and frank psychotic symptoms. Transient perceptual disturbances (TPD) are typically short-lived, mild alterations in perception (visual, auditory, tactile, etc.) that do not indicate a break from reality and are often understandable within a given context, especially in response to fatigue, stress, substance use, sensory deprivation, or sleep deprivation [13]. These are brief, rarely lasting more than a few minutes, and self-limited. Insight is typically preserved: the person may recognize the experience as unusual or not real and they are not associated with disorganized thinking or behavior. Common in this category are hypnagogic or hypnopompic hallucinations, that is, hearing one’s name being called in a quiet room. Complex perceptual disturbances, as in social perception, are typically attributed to neurological defects (such as traumatic brain injury) and are thus non-transitory by nature [13]. Psychotic experiences (PE) are defined as hallucinations/delusions that can occur outside of a psychotic disorder in the general population; they represent a marker of severe psychopathology and a transdiagnostic marker for developing mental disorder [14]. In older adults (age >64), for every 100 older adults, 1 reports an incident of PEs each year contrary to adolescents, where the figure rises from one to five, albeit heavily impacted by higher consumption of psychoactive substances in this age group. Psychotic symptoms, on the other hand, are severe disturbances in perception, thought, and behavior, usually marked by a loss of contact with reality. Frank psychotic symptoms in the elderly include hallucinations, delusions, disorganized thinking/speech, and grossly disorganized or catatonic behavior. Patients typically have a lack of insight regarding the true nature of the symptoms. The prevalence of any psychotic symptom in a large non-demented population has been reported as high as 10.1% [15], and these were associated with a poor prognosis.

Loss of vision in cataract patients evolves over a comparably long period of time and does not reach complete blindness in developed countries; a systematic review and meta-analysis found that maturity for surgery is decided with the best practice guidelines, which relate to the threshold between poor (<20/40) and fair (>20/40) preoperative visual acuity and are specific to local health systems [16]. Preoperative visual acuity did not predict the outcome of cataract surgery, evaluated as both objective and subjective visual improvement. The improvement in visual acuity following successful surgery is typically marked and rapid with eyesight in the operated eye restored within days. Thus, it is plausible to hypothesize that serious loss of vision lasting more than weeks could raise the incidence of psychotic-like symptoms and experiences (paranoid thought, perceptive disturbance) with a chance for full or partial recovery following restoration of vision to a functional level. This is in contrast with psychotic-like symptoms that are a precursor to serious mental illness, which would necessitate appropriate vigilance and, possibly, treatment. Differential diagnosis of those symptoms in the absence of a history of psychosis or cognitive impairment or a distinct neuropsychiatric syndrome (Anton, Charles Bonnet, Capgras, Fragoli) can be attributed to neurodegenerative processes in advance of dementia [17]. Dementia-related psychosis is not uncommon, with roughly half a million cases identified in US Medicare claims during 2013–2018 out of a total of 2,5 million patients with psychosis, with high mortality rates for those patients during one-year and five-year follow-ups [18]. These dire statistics are driving increased awareness of this issue and calling for more research in order to ascertain the neurobiological underpinnings and provide patients with appropriate care [17,19]. Interestingly, vision loss is now considered a potentially modifiable risk factor for dementia [19], with cataract being identified in particular in a related meta-analysis, but not glaucoma or age-related macular degeneration [20]. Thus, pointing out any adverse impact of vision loss in psychotic symptoms could prevent misdiagnosis such as late-onset psychosis or early stages of dementia, reducing the possibility of receiving unnecessary treatment.

The aim of this study is to examine whether there is a relationship between the benefit in visual acuity and decrease in any psychotic symptoms in cataract patients with poor preoperative visual acuity (mature cataract in both eyes) and no prior history of overt psychosis. These patients experienced a non-congenital, significant loss of visual function that developed over a relatively short period of time and was alleviated shortly after surgery. The research hypothesis is that any psychotic symptoms can be alleviated following the successful surgical procedure.

## 2. Materials and Methods

### 2.1. Study Design and Population

This is an observational prospective study of a cohort of 200 consecutive patients who underwent phacoemulsification surgery in the Department of Ophthalmology of the General Hospital of Serres, Greece. The inclusion criteria were poor preoperative visual acuity (<20/40) in both eyes, a waiting time for surgery between two and six months, and willingness to participate in the study following informed consent. The exclusion criteria were any complications related to the phacoemulsification process, previous cataract surgery, the existence of other comorbid eye diseases, any current diagnosis of a psychiatric disorder, and any past diagnosis of a disorder in the psychosis spectrum or severe affective disorder (ICD-10 codes F2x and F3x), including schizophrenia, schizoaffective disorder, bipolar disorder, and delusional disorder. Also, the prescription of medications that may induce psychotic symptomatology, including anticholinergics, benzodiazepines, corticosteroids, dopaminergic agents, was another exclusion criterium. Additionally, illicit substances of abuse, systematic use of benzodiazepines, and systematic use of alcohol higher than moderate, which for men was defined as two units or less in a day and for women one unit or less in a day, were also exclusion criteria. The reported sociodemographic variables were gender, age, and stressful life events in the past six months (such as the death of a loved one or major health issues that necessitated the hospitalization of oneself or of loved ones).

### 2.2. Measures

All patients were initially handed out a brief demographics questionnaire when they were placed on the waiting list, at least a month before surgery. The questionnaire included information about their gender, age, marital status, living arrangements, comorbid health issues, and any life stressors occurring during the past six months, including death or divorce, loss of work or any other self-reported adversities. The Symptoms Checklist 90—Revised (SCL-90-R), a multidimensional self-report symptom inventory [21], and its derived Greek standard version were used in this study [22]. The SCL-90-R is a multidimensional self-report measure assessing the breadth and severity of current psychological symptoms and distress, consisting 90 Likert-type questions divided into 9 symptom dimensions. Two symptom dimensions were chosen for this study, namely Paranoid Ideation and Psychoticism. Paranoid ideation is depicted in this scale as a disordered way of thinking. Projective thinking, hostility, suspiciousness, grandiosity, centrality, fear of losing autonomy, and delusions are considered primary manifestations of this disorder. The concept of Psychoticism is described here as a continuous spectrum of alternate human experience with a range from mild interpersonal alienation to significant signs of psychosis. The items included withdrawal, isolation, and a schizoid lifestyle, along with first-rank schizophrenia symptoms such as hallucinations and thought-broadcasting. The items of Psychoticism and Paranoid Ideation are presented in Table 1.

Following the baseline measurement, a second measurement was carried out with the SCL-90-R two months after surgery, during the routine post-surgery assessment. Additionally, the patients’ best-corrected visual acuity (BCVA) for the affected eye was also measured pre- and post-surgery using the Early Treatment Diabetic Retinopathy Study (ETDRS) charts.

### 2.3. Statistics

Gender differences on age and the Paranoid Ideation and Psychoticism scores were assessed with Mann–Whitney tests. The differences in the Paranoid Ideation and Psychoticism scores pre- and post-operation were assessed using the paired-samples Wilcoxon test. Two stepwise regression analyses were conducted to assess the impact of visual acuity on the difference in Paranoid Ideation and Psychoticism scores pre- and post-operation while controlling for gender, age, and stressful life events. All statistics were calculated using the SPSS statistical package, version 26 [23].

## 3. Results

The sample demographics are presented in Table 2. There were 150 patients, including 86 men (57.3%) and 64 women (42.7%). The mean age for men was 73.84 years (SD = 8.55 years), and for women, 73.45 years (SD = 7.055); the difference was not statistically significant. The preoperative mean SCL-90-R scores were 6.42 (SD = 5.248) for men and 8.13 (SD = 5.683) for women. The postoperative mean SCL-90-R scores were 5.43 (SD = 3.933) for men and 6.72 (SD = 4.732) for women.

Men scored lower than women on the SCL-90-R Paranoid Ideation scale preoperatively (Mann–Whitney Z = 2.013, *p* = 0.044) and postoperatively (Mann–Whitney Z = 2.191, *p* = 0.028). The scores on the Psychoticism scale did not differ statistically significantly preoperatively (Mann–Whitney Z = 1.762, *p* = 0.078) or postoperatively (Mann–Whitney Z = 0.728, *p* = 0.467). A paired-samples Wilcoxon test revealed statistically significant difference in the preoperative and postoperative Paranoid Ideation and Psychoticism scores, Wilcoxon Z = 5.425, *p* < 0.001 and Wilcoxon Z = 6.478, *p* < 0.001, respectively. Effect size d was 0.383 for the Paranoid Ideation postoperative difference and 0.485 for the Psychoticism postoperative difference, denoting medium strength of the relationship between the outcome of the surgery and drop in Paranoid Ideation and Psychoticism scores [24]. The difference in Paranoid Ideation and Psychoticism scores pre- and postoperatively correlated with the difference in BCVA pre- and postoperatively (Spearman r_s_ = 0.172, *p* = 0.015 for the Paranoid Ideation scale and Spearman r_s_ = 0.49, *p* < 0.001 for the Psychoticism scale).

An assessment was carried out on the pre- and postoperative difference in each separate item of the Paranoid Ideation and Psychoticism scales. In the case of the Paranoid Ideation scale items, target *p* for an alpha of 0.05 was set at <0.008 to adjust for multiple comparisons using the Bonferroni criterion. Three items met this threshold, namely item 8 (‘feeling others are to blame for most of your troubles’), item 18 (‘feeling that most people cannot be trusted’), and item 83 (‘feeling that people will take advantage of you if you let them’). These were the items on the SCL-90-R that contributed the most to the postoperative improvement. The similar process for the Psychoticism scale had target *p* for an alpha of 0.05 set at <0.005. Four items met that threshold, namely item 16 (‘hearing words that others do not hear’), 62 (‘having thoughts that are not your own’), 84 (‘having thoughts about sex that bother you a lot’), and 85 (‘the idea that you should be punished for your sins’).

Two separate stepwise regression analyses were conducted to assess the impact of selected sociodemographic factors and the improvement in visual activity on the Paranoid Ideation and Psychoticism scores. The variables that were entered in the analyses were sex, age, the existence of stressful life events during the past six months and the difference in visual acuity scores pre- and postoperatively. Table 2 presents the results from the stepwise regression of the Paranoid Ideation scores; F (3, 196) = 17.254, *p* < 0.001, adjusted r square = 0.197. The variables that were retained in the final model were age, stressful life events and difference in visual acuity, with lower age, more stressful live events preoperatively and higher improvement in visual acuity being associated with higher improvement in Paranoid Ideation scores (Table 3).

Table 4 presents the results from the stepwise regression of the Psychoticism scores; F (2, 196) = 43.146, *p* < 0.001, adjusted r square = 0.299. The variables that were retained in the final model were stressful life events and difference in visual acuity, with more stressful live events preoperatively and higher improvement in visual acuity being associated with higher improvement in the Psychoticism scores (Table 2).

## 4. Discussion

Our results indicate an improvement in paranoid and psychotic symptomatology following cataract surgery, in accordance with the research hypothesis. The effect size for the relationship between change in BCVA and psychosis-related symptomatology was moderate, denoting that these results can be meaningful in everyday life. To-date this is the first study examining the positive impact of the improvement of visual acuity in psychosis-related symptomatology in patients with no prior history of psychosis. The implications are that loss of visual function related to cataract carries an additional risk for psychotic symptomatology, which may be problematic for patients who already present with psychotic disorders. Of special note is that the cataract patient is typically over 65 years of age and may show early signs of age-related dementia; the decrease in visual acuity may be an additive factor to psychotic symptoms induced by cognitive decline.

The initial differences between the genders on Paranoid Ideation and Psychoticism scores are in line with published research [25]. However, in our sample, the scores on Psychoticism were notably higher than those of Paranoid Ideation. This may suggest a more acute stressor on psychic stability, since Paranoid Ideation items refer to more constant features of one’s psychic apparatus, such as distrust of others, whereas Psychoticism items correspond directly to symptoms of overt psychosis, such as hallucinations and psychotic guilt.

We should note that Paranoid Ideation and Psychoticism symptoms were assessed using a questionnaire and not a psychiatric interview; however, none of the participants had a history of psychosis or dementia, and they were not under antipsychotic medication or medication that may induce psychotic symptoms. Follow-up questions in a psychiatric interview could lead to a clearer distinction between transient perceptual disturbances, prodromal symptoms, and psychotic symptoms within the patient sample. However, the study design with a follow-up on each patient ensures that the improvement on those symptoms is seen consistently with how each patient perceives his/her experiences before and after surgery. The two relevant categories examined in the SCL-90-R, Paranoid Ideation and Psychoticism, contain only a single item that could be related to TPD, namely item 16 (‘Hearing voices that other people do not hear’) (Table 1). The rest of the symptoms in the Psychoticism and Paranoid Ideation scales have been categorized as prodromal symptoms of psychosis. These are both positive and negative symptoms, and they include, among others, subthreshold or attenuated suspiciousness or paranoia, ideas of reference, magical thinking, somatic delusions, social withdrawal, or reduced interest in relationships.

A limitation of the study is its observational design, which cannot infer causality. However, the short timeframe of the research reduces the possibility that the alleviation of the psychotic symptoms can be attributed to other external causes within only a month. Improvement of general mental health due to postoperative relief is a valid possibility, although it would be expected to impact mostly anxiety and depression. A related confounder for our study is that any surgical procedure involves stress, which could have affected preoperative scores negatively and when the surgical procedure was carried out and stress subsided then the scores on Psychoticism and Paranoid Ideation were also reduced. Cataract surgery is a minimally invasive surgery and the patient typically returns to his/her home within a few hours. A past study of psychological distress in hospitalized patients [26] concluded that the Paranoid Ideation scores of hospitalized patients were similar to non-patients, and the Psychoticism scores were not statistically significantly affected by hospitalization. Hence, the notion that the mere prospect of receiving cataract surgery could increase those scores preoperatively is unlikely.

## 5. Conclusions

Our findings indicate that improvement of psychotic-like symptoms following phacoemulsification surgery is associated with an improvement of visual acuity. The implications are important, since the timely resolution of visual defects of the elderly could be protective of their mental stability, even more so in cases of pre-existing psychotic disorder or severe cognitive decline. Patients presenting with unexplained psychotic symptoms in the backdrop of severe visual loss should be offered an accelerated pathway to treatment. Such results are likely to be replicated with patients presenting with other visual defects that result in significant vision loss.

## Figures and Tables

**Table 1 diseases-13-00224-t001:** Items on the Psychoticism and Paranoid Ideation SCL-90-R scale [21].

	**Psychoticism**
**Item #**	**Wording**
7	The idea that someone else can control your thoughts
16	Hearing voices that other people do not hear
35	Other people being aware of your private thoughts
62	Having thoughts that are not your own
77	Feeling lonely when you are with people
84	Having thoughts about sex that bother you a lot
85	The idea that you should be punished for your sins
87	The idea that something serious is wrong with your body
88	Never feeling close to another person
90	The idea that something is wrong with your mind
	**Paranoid Ideation**
**Item #**	**Wording**
8	Feeling that others are to blame for most of your troubles
18	Feeling that most people cannot be trusted
43	Feeling that you are watched or talked about by others
68	Having ideas or beliefs that others do not share
76	Others not giving you proper credit for achievements
83	Feeling that people will take advantage of you if you let them

**Table 2 diseases-13-00224-t002:** Sample demographics and SCL-90-R scores by gender.

	Male	Female
Participants	115 (57.5%)	85 (42.5%)
Age (mean/SD)	73.92 (8.46)	73.51 (6.917)
Family status		
Married	77 (67%)	41 (48.2%)
Widowed	29 (25.2%)	39 (45.9%)
In a relationship	8 (7%)	3 (3.5%)
Without a relationship	1 (0.9%)	2 (2.4%)
Stay		
Alone	24 (20.9%)	34 (40%)
With family/partner	91 (79.1%)	51 (60%)
Require a caretaker		
No	92 (80%)	56 (65.9%)
Yes	23 (20%)	29 (34.1%)
Recent (<6 months) stressful event (loss of a loved one, serious health issue)		
No	101 (87.8%)	75 (88.2%)
Yes	14 (12.2%)	10 (11.8%)
Preoperative Paranoid Ideation score	0.214 (0.28)	0.292 (0.33)
Postoperative Paranoid Ideation score	0.139 (0.17)	0.215 (0.25)
Preoperative Psychoticism score	0.274 (0.26)	0.355 (0.3)
Postoperative Psychoticism score	0.221 (0.2)	0.255 (0.24)

**Table 3 diseases-13-00224-t003:** Stepwise regression on the difference in the Paranoid Ideation scores pre- and postoperatively as the dependent variable.

Model	Standardized Beta	t	*p*	95.0% Confidence Interval for B	
				Lower Bound	Upper Bound
(Constant)		−1.763	0.080	−0.025	0.450
Age	−0.134	−2.085	0.038	−0.007	0.000
Stressful life event <6 months ago	0.369	5.751	<0.001	0.341	0.697
Difference in visual acuity pre- and postoperatively	0.250	3.909	<0.001	0.176	0.535

**Table 4 diseases-13-00224-t004:** Stepwise regression on the difference in the psychoticism scores pre- and postoperatively as the dependent variable.

Model	Standardized Beta	t	*p*	95.0% Confidence Interval for B	
				Lower Bound	Upper Bound
(Constant)		−4.349	<0.001	−0.129	−0.049
Stressful life event <6 months ago	0.187	3.131	0.002	0.073	0.322
Difference in visual acuity pre- and postoperatively	0.509	8.527	<0.001	0.419	0.672

## Data Availability

The raw data supporting the conclusions of this article will be made available by the authors on request.

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
