# Peer review of "Psychotic Symptoms in Cataract Patients Without Overt Psychosis Are Ameliorated Following Successful Cataract Surgery"

_diseases, 2025, doi:10.3390/diseases13070224_

Round 1
Reviewer 1 Report
Comments and Suggestions for Authors
Dear authors,
I commend your efforts in analyzing a significant number of patients. However, I have to point out some criticisms: since your work is not the first on the same topic, I think you could have avoided the potential shortcomings of the study.
Please explain how long before surgery you tested the patients. I think it is not a good idea to test patients immediately before surgery and that the impact of the stress of surgery could have been avoided if the patients had been tested a few days before surgery.
Author Response
Comments: I commend your efforts in analyzing a significant number of patients. However, I have to point out some criticisms: since your work is not the first on the same topic, I think you could have avoided the potential shortcomings of the study.
Please explain how long before surgery you tested the patients. I think it is not a good idea to test patients immediately before surgery and that the impact of the stress of surgery could have been avoided if the patients had been tested a few days before surgery.
Response:
Dear anonymous reviewer. Thank you for your kind words and for pointing out this detail which was not explained properly in the original manuscript. The patients were tested with the first questionnaire not during the final pre-operative assessment which is carried out a day before surgery but when the patient was confirmed on the waiting list for surgery, which was as mentioned in the original manuscript at least one month before surgery. This has been made clearer in the revised manuscript.
Also, the manuscript has been enlarged considerably with additional references in the Introduction section on previous work and in the Discussion section following your feedback.
Reviewer 2 Report
Comments and Suggestions for Authors
This manuscript explores a relevant but under-investigated topic: the relationship between visual restoration following cataract surgery and improvement in psychosis-related symptoms. The study is clearly structured and addresses an original hypothesis with potential clinical relevance, particularly for elderly populations. However, there are several concerns that limit its current scientific robustness. First, the use of self-reported questionnaires (SCL-90-R) without clinical validation or corroborating psychiatric interviews undermines the reliability of the psychotic symptom measurements. Additionally, although the authors control for basic demographic and stress-related variables, the regression models lack more nuanced psychiatric or cognitive assessments, which could better distinguish subclinical psychotic features from normal aging or anxiety-related distress. The manuscript would benefit from a clearer distinction between transient perceptual disturbances and psychotic symptoms. Moreover, the writing occasionally lacks precision and contains grammatical inconsistencies. Finally, while the findings are intriguing, causality cannot be inferred from the observational design, and the discussion does not adequately address alternative explanations or confounding variables (e.g., placebo effect, postoperative relief).
Author Response
Comments 1. This manuscript explores a relevant but under-investigated topic: the relationship between visual restoration following cataract surgery and improvement in psychosis-related symptoms. The study is clearly structured and addresses an original hypothesis with potential clinical relevance, particularly for elderly populations.
Response 1: Dear anonymous reviewer. Thank you for your kind words. Your comments have led to a considerable improvement of the manuscript, with new sections added in the Introduction and the Discussion sections.
Comment 2: However, there are several concerns that limit its current scientific robustness. First, the use of self-reported questionnaires (SCL-90-R) without clinical validation or corroborating psychiatric interviews undermines the reliability of the psychotic symptom measurements.
Response 2: Thank you for your comment, this has been incorporated in the revised section on limitations of the study.
Comment 3: Additionally, although the authors control for basic demographic and stress-related variables, the regression models lack more nuanced psychiatric or cognitive assessments, which could better distinguish subclinical psychotic features from normal aging or anxiety-related distress. The manuscript would benefit from a clearer distinction between transient perceptual disturbances and psychotic symptoms.
Response 3: Thank you for your comment which has led to a considerable improvement in the revised manuscript with extensive sections dedicated to the distinction between different types of psychotic symptoms.
Comment 4: Moreover, the writing occasionally lacks precision and contains grammatical inconsistencies.
Response 4: Thank you for your comment, the manuscript has been checked for consistency in the definitions that are employed throughout
Comment 5: Finally, while the findings are intriguing, causality cannot be inferred from the observational design, and the discussion does not adequately address alternative explanations or confounding variables (e.g., placebo effect, postoperative relief).
Response 5: Thank you for your comment, this has been incorporated in the discussion section and the revised section on limitations of the study.
Round 2
Reviewer 2 Report
Comments and Suggestions for Authors
Accept in the current form